# Whole-Genome Resequencing in Sheep: Applications in Breeding, Evolution, and Conservation

**DOI:** 10.3390/genes16040363

**Published:** 2025-03-22

**Authors:** Ruoshan Ma, Ying Lu, Mengfei Li, Zhendong Gao, Dongfang Li, Yuyang Gao, Weidong Deng, Bo Wang

**Affiliations:** 1Yunnan Provincial Key Laboratory of Animal Nutrition and Feed, Faculty of Animal Science and Technology, Yunnan Agricultural University, Kunming 650201, China; maruoshan_2000@163.com (R.M.); yinglu_1998@163.com (Y.L.); mfli_2000@163.com (M.L.); zander_gao@163.com (Z.G.); dfli0927@163.com (D.L.); gaoyy5210@163.com (Y.G.); 2State Key Laboratory for Conservation and Utilization of Bio-Resource in Yunnan, Kunming 650201, China

**Keywords:** whole-genome resequencing, sheep, genetic map, population genetic differentiation, trait association analysis, genetic resources

## Abstract

Sheep (*Ovis aries*) were domesticated around 10,000 years ago and have since become an integral part of human agriculture, providing essential resources, such as wool, meat, and milk. Over the past century, advances in communication and agricultural productivity have driven the evolution of selective breeding practices, further enhancing the value of sheep in the global economy. Recently, the rapid development of whole-genome resequencing (WGR) technologies has significantly accelerated research in sheep molecular biology, facilitating the discovery of genetic underpinnings for critical traits. This review offers a comprehensive overview of the evolution of whole-genome resequencing and its application to sheep genetics. It explores the domestication and genetic origins of sheep, examines the genetic structure and differentiation of various sheep populations, and discusses the use of WGR in the development of genetic maps. In particular, the review highlights how WGR technology has advanced our understanding of key traits, such as wool production, lactation, reproductive performance, disease resistance, and environmental adaptability. The review also covers the use of WGR technology in the conservation and sustainable utilization of sheep genetic resources, offering valuable insights for future breeding programs aimed at enhancing the genetic diversity and resilience of sheep populations.

## 1. Introduction

There are over 1000 sheep breeds worldwide, and sheep are one of the most important livestock species for farmers and herders in developing countries. According to the latest data from the Food and Agriculture Organization (FAO, http://www.fao.org/faostat/en/#data/QA, accessed on 11 November 2024), the global sheep population reached 1.5 billion in 2022. These sheep are widely distributed across more than 170 countries and regions, particularly in developing countries, such as China and India, where they play a crucial role in providing essential resources, such as meat, milk, skin, and wool fibers (Figure 1).

Sheep are among the earliest domesticated animals by humans [1]. Archaeological records suggest that domestication first occurred in Southwest Asia. Although many aspects of the domestication process, such as the specific location, timing, and post-domestication history, remain unresolved [2]. Sheep have developed genetic variations with ideal phenotypic effects over the course of evolution, making them an ideal model for biomedical research. Moreover, sheep play a crucial role in providing protein to large populations worldwide [3]. While molecular biology research and genetic breeding efforts for sheep have lagged behind those for economically more significant livestock species, such as cattle and pigs [4], the continuous advancement and application of sequencing technologies have led to an increasing number of studies on sheep genetic breeding [5,6,7].

By deeply comparing and analyzing genomic data from different genetic resources, researchers can link specific genes to traits, identifying genetic resources of particular value. This approach, leveraging modern biotechnology, especially advancements in genomics and molecular biology, provides new avenues for the precise identification and improvement of genetic resources [8]. It not only aids in the accurate identification and transformation of local genetic resources but also promotes the continued development and application of desirable traits, offering scientific evidence for genetic improvement and contributing to the breeding of sheep varieties with better adaptability to specific environments and higher production performance.

## 2. Whole-Genome Resequencing

Whole-genome resequencing enables high-throughput sequencing and alignment of individuals with known genomes, uncovering molecular-level genetic characteristics. This provides a powerful tool for predicting candidate genes for important economic traits in animals and conducting genetic evolution analyses [9]. The technique is highly favored due to its ability to detect a wide range of variation types, its precise and efficient operation, cost-effectiveness, and broad applicability [10,11,12]. These genetic variations form the basis for understanding the genetic architecture of economically important traits in sheep, which directly benefits genomic selection and breeding programs.

With the continuous advancement of sequencing technologies and decreasing costs, whole-genome resequencing has become an indispensable tool for not only characterizing genetic diversity and adaptive evolution in sheep populations but also accelerating gnomic selection by enabling the identification of markers linked to economically important traits. This technology not only enables the screening and analysis of large-scale genetic variation data through bioinformatics methods, including single nucleotide polymorphisms (SNPs), insertions and deletions (InDels), structural variations (SVs), and copy number variations (CNVs) [13], but also enriches the existing genomic sequence resources, providing new perspectives for understanding biological genetic diversity and evolutionary mechanisms [14]. Whole-genome resequencing technology has undergone three major innovations, resulting in different sequencing technologies [15]. From the first-generation Sanger sequencing to the third-generation single-molecule real-time sequencing and nanopore sequencing, each generation of technology has its unique advantages and disadvantages [16], as shown in Figure 2.

In studies of population genetic differentiation, whole-genome resequencing technology has helped scientists understand genetic differences between populations, providing crucial data support for biodiversity conservation and species evolution research [17,18,19]. Unlike whole-genome sequencing, which involves sequencing a genome for the first time to create a reference sequence, whole-genome resequencing focuses on sequencing multiple individuals within a species and aligning their sequences to an existing reference genome [20,21]. In genetic mapping, this technology has made it possible to construct high-density genetic maps, significantly enhancing the accuracy of gene localization [22]. In trait association analysis, by linking genotype and phenotype data, whole-genome resequencing has revealed the genetic basis of complex traits, offering scientific evidence for breeding improvements [23]. Moreover, this technology plays an important role in the utilization and conservation of genetic resources by identifying and protecting valuable genetic variations, supporting the sustainable use of these resources [24]. In summary, whole-genome resequencing technology is of great significance in fields, such as population evolution, trait gene function, disease diagnosis and treatment, and genetic resource conservation, providing strong support for future research in sheep genetic breeding and resource protection.

## 3. Origin and Evolution of Sheep

Domesticated sheep are derived from the wild Asiatic mouflon of Southwest Asia [25], having become one of the earliest domesticated livestock species around 10,000 BCE, alongside goats [26]. As human migration spread, domesticated sheep were widely disseminated, playing a key role in both settled agriculture [27] and nomadic societies [28]. Domesticated European sheep mainly entered present-day Europe via inland routes across the Mediterranean and Northern Europe. The Mongolian Plateau, located in the northeastern region of the Eurasian continent, served as an important “transport hub”. Its unique geographical position and natural environment made it a crucial passage for ancient East–West cultural exchanges and a key point in the dissemination of sheep domestication techniques [29]. In addition to the initial expansion, archaeological records also reveal the spread of other populations, such as the snow sheep and its American relatives (bighorn sheep and Dall sheep), indicating their penetration into regions, such as Asia Minor, Asia, and the European mouflon [30]. Additionally, the adaptive penetration of wild sheep into the Asian and European mouflon populations is considered an independent event [31]. Furthermore, snow sheep and bighorn sheep shared some overlapping regions before or during the domestication process. Subsequent studies suggest that the now-extinct European sheep may have experienced infiltration through interbreeding with wild domesticated sheep around 6000–5000 BCE [32]. The origin of local sheep populations in Morocco remains unclear. In addition to the long-established mountain populations, other local breeds may have been introduced by the Phoenicians, Romans, and Arabs or brought from regions south of the Sahara [33].

The evolutionary history of the genus *Ovis* spans 8.31 million years and includes eight extant species: domestic sheep, bighorn sheep (*Ovis ammon*), Asiatic mounflon (*Ovis orientalis*), European mouflon (*Ovis musimon*), urial (*Ovis vignei*), bighorn sheep (*Ovis canadensis*), thinhorn sheep (*Ovis dalli*), and snow sheep (*Ovis nivicola*) [32,34]. Due to the complexity of the environmental conditions in which they reside, local populations have undergone significant differentiation in appearance, size, growth performance, and reproductive traits under the continuous drive of selective pressures [35]. By analyzing the pathways of introgression in 154 domesticated breeds and 7 wild relatives of the 8 existing sheep species, researchers have further illustrated the introgression routes of sheep [36].

The wide array of domesticated sheep breeds and their diverse phenotypes is a testament to centuries of both natural and artificial selection, encompassing enhanced breeds, native Chinese varieties, African breeds, and their wild Asian mouflon forebears [37]. In terms of population structure, domesticated sheep collectively form a single evolutionary lineage, whereas Chinese indigenous sheep clearly reflect geographical distribution patterns [38]. Across various livestock species, this diversification has been traced to distinct migration pathways at different historical junctures, leading to the emergence of numerous breeds finely tuned to their local environments [39]. This indicates that the domestication of sheep was a widespread phenomenon, incorporating diverse maternal lineages. Through long-term natural and artificial directional selection, domesticated livestock have gradually developed significant genetic differences in phenotypic traits, important economic characteristics, and environmental adaptability, greatly enriching the diversity of existing biological genetic resources [40].

## 4. Construction of Sheep Genetic Maps

In summary, analytical methods for studying population structure elucidate genetic relationships and ancestral histories, enhancing our understanding of genetic variation and species evolution. With the rapid development of molecular biology and genomics, the types of genetic maps have continuously increased, and the density of markers has become progressively higher, making the construction of high-precision genetic linkage maps a popular research topic. A genetic linkage map represents the relative positions of genes and specific polymorphic markers within a genome. The process of constructing such a map primarily involves obtaining polymorphic markers from animals, establishing reference or resource families, performing genotype analysis on polymorphic loci in these families, and ultimately constructing the genetic map [41,42]. The development of high-resolution genetic maps has facilitated trait association studies, providing a foundation for linking genetic variations with economically important traits in sheep. The completion of high-quality sheep genome sequence maps and functional annotations has provided an excellent reference standard for sheep genomics research, allowing researchers to more accurately identify and analyze genetic variations within the sheep genome, thereby deepening the understanding of sheep’s genetic structure and function [43]. These research findings not only help reveal the genetic basis of complex traits in sheep but also provide scientific evidence for genetic improvement and disease prevention in sheep.

The first autosomal linkage map of sheep was published in 1995. This map, constructed through linkage analysis of unknown microsatellite markers in the genomes of sheep, cattle, and mice, revealed a close relationship between the sheep linkage map and those of cattle and goats [44]. Three years later, de Gortari et al. [45] developed the second-generation sheep genetic map, which improved the density and accuracy of genetic markers, providing strong support for more detailed genetic analysis and breeding. In 2001, Jillian F. Maddox et al. [46] published the third-generation sheep genetic linkage map, covering a broad range of the sheep genome. By analyzing 246 polymorphic markers, they established a genetic linkage map using nine full sibling families across three generations.

In 2014, Jiang et al. [47] sequenced and assembled the reference genome of the Tsiang sheep. In 2021, researchers employed resequencing techniques to examine the distribution of whole-genome copy number variations (CNVs) across 32 fine wool sheep from 3 distinct breeds. This study resulted in the creation of a CNV map specific to the genomes of Chinese indigenous fine wool sheep. The findings offered significant genetic variation data for sheep genomics and contributed valuable knowledge for the investigation of complex traits in sheep [48]. In 2022, the research team developed the first visualized sheep pan-genome map, revealing structural variations and their relationship with sheep traits [49]. In 2025, the first complete reference genome of sheep, from telomere to telomere (T2T-sheep1.0), was successfully assembled [50]. This genome identified many previously unresolved regions, corrected structural errors in earlier reference assemblies, and improved the ability to detect structural variations in repetitive sequences. Overall, there is significant genetic variation within sheep and their wild relatives, providing a foundation for molecular breeding, but their genome maps have not been fully explored [51]. Over the past two decades, advancements in sequencing technologies have led to the continuous emergence of more detailed genomic maps [52], whole-genome resequencing has been confirmed as a key tool for exploring genetic variation, species adaptive evolution, genetic diversity, and the screening of candidate genes for key traits in livestock and poultry [53]. This technology allows for a more accurate understanding of the true composition of populations, revealing important issues, such as the geographical origin, evolutionary history, and population stratification.

## 5. Genetic Structure Differentiation in Sheep Populations

Population genetic structure refers to the non-random distribution pattern of genomic genetic variation within a species or population. This distribution pattern reflects the genetic differences and phylogenetic relationships both within and between populations [54]. Within the same population, different individuals often exhibit relatively high relatedness due to shared genetic backgrounds and inbreeding, thereby sharing more genetic variation. These individuals show greater genetic similarity, resulting in relatively low genetic diversity within the population. Genetic diversity and its whole-genome variation are influenced by various factors, including effective population size, population structure, inbreeding, and migration [55]. Through whole-genome analysis of sheep, researchers can determine the genetic diversity, population structure, and selective traits of the population [56,57], providing scientific evidence for breeding improvement measures for local sheep genetic resources. Moreover, whole-genome resequencing aids in population genetic structure analysis, helping researchers infer the number of populations, assign individual identities, and identify migratory individuals and the continuity of genetic resources [58], thus contributing to the understanding of their evolutionary history.

Utilizing population genome data, the research team conducted whole-genome resequencing on 10 individual Luxi Blackhead sheep. Among the diverse sheep breeds native to China are the Black Head Dorper and the Small Tailed Han sheep [59]. Research has revealed that the LuXi Blackhead sheep constitutes a unique population, setting it apart from known breeds and surpassing other indigenous Chinese sheep in meat quality. A separate investigation into 13 native British breeds and 6 hybrid breeds (BAR) demonstrated that commercial breeds generally show increased admixture, a less defined population structure, and higher heterozygosity in comparison to the more distinct native breeds. These native breeds, on the other hand, typically exhibit less admixture, a narrower genetic diversity, and elevated coefficients of relatedness [60]. The study revealed that Liangshan semi-fine-wool sheep exhibit greater genetic similarity to Border Leicester and Romney sheep compared to their resemblance to Tibetan sheep, Yunnan sheep, and Chinese Merino sheep [61]. Furthermore, all individuals of Liangshan semi-fine-wool, White Suffolk, and Butuo Black sheep clustered together, clearly distinguishing these breeds, while Jialuo and Mage sheep exhibited the closest genetic relationships, making them difficult to differentiate [62].

## 6. Sheep Trait Association Analysis

Sheep play a crucial role in animal husbandry, providing not only meat and dairy products but also wool, making them a significant source of economic income for farmers and breeders [63]. To improve sheep quality and production efficiency, researchers have utilized whole-genome resequencing technology to investigate the genetic basis of traits such as wool quality, milk production, and reproductive performance, as shown in Figure 3. The application of this technology, on the one hand, facilitates the breeding of superior sheep breeds, and on the other, enhances specific traits through genetic improvement, thereby improving overall quality and productivity. This genomic knowledge provides breeders with new tools to accelerate genetic improvement through marker-assisted selection and genomic selection, thus improving breeding efficiency and achieving balanced selection for multiple traits. Furthermore, genomic sequencing technology is of great importance in exploring the genetic mechanisms of livestock diseases. By analyzing the sheep genome, scientists can accurately identify genetic markers associated with diseases, which can then be used to develop effective prevention and control strategies [64].

### 6.1. Wool Fineness and Color

Wool is an ideal raw material for producing high-quality textiles and apparel, with significant economic value. The wool industry contributes substantially to the global economy, generating considerable income for numerous countries and regions. Wool fineness is one of the key indicators of wool quality, directly influencing the feel and comfort of textiles. The finer the wool, the thinner the fibers, and the softer and more comfortable the resulting fabric. Zhang et al. [65] conducted whole-genome resequencing of 8 sheep breeds and identified candidate genes associated with wool fineness, such as *LGR4*, *PIK3CA*, and *SEMA3C*. Additionally, a study involving the resequencing of 460 sheep from 4 distinct fine-wool breeds identified a correlation between genes, such as *RHPN2* and *NRXN1*, which were associated with wool fineness [66]. These studies reveal genomic information related to important sheep traits and identify valuable candidate genes; however, these genes often only account for a certain proportion of wool thickness variation, and there is limited research on related genes, such as those involved in fiber diameter and follicle development.

The diversity of wool color not only enriches the variety of textile products but also meets the market demand for wool in different colors. A deeper understanding of the genetic basis of wool color allows scientists to identify key genes associated with pigmentation, thereby providing breeders with scientific evidence to develop effective breeding strategies to enhance or alter wool color. Cao et al. [67] analyzed the population structure, genetic diversity, and selection traits of Romanov sheep using whole-genome sequencing data from 114 individuals, including 17 Romanov sheep and 10 other European breeds, and identified *MC1R* as a candidate gene associated with the breed’s unique gray wool color. A study was carried out involving whole-genome resequencing of 48 sheep, achieving an average sequencing coverage of 7.6×. This research led to the identification of several genes that are instrumental in the development and control of wool color [68]. These include *PDE4B*, *GMDS*, *RCOR1*, and *TECRL*, which are involved in melanocyte stem cell regeneration, melanocyte proliferation and differentiation, melanin synthesis and distribution, and color variation, affecting the formation of non-white wool phenotypes. Genes, such as *ABCD4*, *VSX2*, *ITCH*, *NNT*, *POLA1*, and *DAO*, are involved in inhibiting melanocyte pigment deposition, proliferation, and migration, thereby affecting the formation of white wool phenotypes.

### 6.2. Lactation Traits

Lactation traits are among the most important economic traits in mammals, as the amount of milk produced directly determines the survival rate and growth rate of offspring [69,70]. Although the population size of dairy sheep is smaller than that of meat or wool sheep and research on dairy sheep is relatively limited, the lactation mechanism of dairy sheep is crucial for improving animal production methods. Through whole-genome resequencing, candidate genes associated with lactation traits can be effectively identified and validated, revealing genetic variations that affect lactation traits. These variations can serve as molecular markers for subsequent breeding and genetic improvement.

An investigation [71] was conducted on 57 high-yielding dairy sheep and 44 low-yielding dairy sheep, obtaining a large number of effective SNPs across 10 sheep breeds. They conducted population genetic structure analysis, gene detection analysis, and gene functional validation, finding a significant negative correlation between *FCGR3A* and milk production in sheep. Furthermore, a comparative analysis was performed on the genomes of Assaf and Awassi sheep breeds in relation to the Cambridge, Romanov, and British du cher sheep. This research highlighted key genes significantly associated with lactation traits, including *ST3GAL1*, *CSN1S1*, *CSN2*, *OSBPL8*, *SLC35A3*, *VPS13B*, *DPY19L1*, *CCDC152*, *NT5DC1*, *P4HTM*, *CYTH4*, *METRNL*, *U1*, *U6*, and *5S_RRNA* [72]. Through genomic analysis of 46 East Friesian sheep, Li et al. [73] finally identified 4 important KEGG enrichment pathways and 10 candidate genes (*SMOX*, *HIBCH*, *GLI2*, *TXNRD3*, *TRAF3*, *FGF16*, *TRAF3*, *PSMD13*, *SIN3A*, *MRE11*) that were closely associated with milk production in East Friesian sheep.

### 6.3. Reproductive Traits

Reproductive traits in sheep directly impact the economic efficiency of the sheep industry [74]. For example, the lambing rate and ovulation rate are key indicators of sheep reproductive performance, and their levels are directly related to farm production efficiency and economic outcomes [75]. The number of lambs born is a direct reflection of reproductive capability, and the quality of reproductive performance largely determines the overall economic profitability of the farm [76]. Through sequencing the *BMP15* gene in Mongolian sheep, researchers identified 12 novel variants using direct sequencing and whole-genome resequencing techniques. These variants could potentially act as beneficial genetic markers for improving lambing rates [77]. Additionally, a study [78] was conducted involving whole-genome resequencing of four prominent sheep breeds from northwest China, namely Hu, Suffolk, Kazakh, and Duolang sheep, and identified *PAK1*, *CYP19A1*, and PER1 as potential candidate genes responsible for seasonal reproduction in local sheep. Bao et al. [79] used a repeatability-based GWAS analysis and observed two significant SNPs associated with litter size in Hu sheep. By integrating selection traits and GWAS results, they identified 15 candidate genes related to litter size, with *BMPR1B* and *UNC5C* being particularly noteworthy.

Increasing the production capacity of sheep has long been a key issue that experts have focused on addressing. The number of ovulations in sheep directly affects their production efficiency, as a higher ovulation rate theoretically leads to a greater number of lambs born, which is closely tied to the overall productivity and economic profitability of the flock [80]. This directly impacts the flock’s overall production and economic benefits. A study on the genetic variants of Noire du Velay (NV) sheep using whole-genome resequencing and identified a novel SNP located on the *BMP15* gene that is significantly associated with lambing rate. This variant influences *BMP15* expression in oocytes, thereby regulating ovulation rate in sheep [81]. Furthermore, A comparative analysis of genomic selection traits between Hu sheep and various other breeds pinpointed genes, like *BMPR1B* and *PPM1K*, which are notably linked to high reproductive performance in sheep [82].

### 6.4. Multi-Vertebrae Features

The vertebral column provides animals with basic structural and protective functions, and different species possess unique total numbers of vertebrae [83]. In general, sheep have six lumbar vertebrae and five thoracic vertebrae. However, studies have revealed some vertebral variations in domestic sheep, including the phenomenon of multiple vertebrae. High-throughput whole-genome resequencing of sheep with multiple lumbar vertebrae and normal lambs identified *SFRP4* as a potential key gene affecting the number of lumbar vertebrae in sheep [84], and it may be used for sheep breeding in the future. Subsequently, researchers [85] collected 400 sheep with increased thoracic vertebrae and 200 normal Kazakh sheep, generating and sequencing 60 genomic DNA libraries. The results showed that sheep with a higher number of thoracic vertebrae exhibited significantly higher expression levels of the *VRTN* gene during fetal development compared to those with normal thoracic vertebrae counts.

The trait of multiple vertebrae is a beneficial mutation that can significantly increase meat production. An increase in the number of vertebrae leads to a longer body, enhanced meat and wool yield, greater skin area, improved adaptability, and better feed efficiency [86]. Therefore, using whole-genome resequencing to analyze the multiple vertebrae trait in domestic sheep is of great significance for the conservation of local breeds and the improvement of economic benefits.

### 6.5. Ear Shape and Horn Shape

Ear morphology is one of the key traits of sheep breeds, encompassing various variables such as ear size and position, all of which are controlled by multiple genes. By analyzing 34k SNP genotyping data, researchers identified genomic regions associated with ear morphology in a sample of 515 sheep from 17 different breeds. The findings indicate that the genetic basis for large, soft ears is possibly situated within a 175 kb interval downstream of the *MSRB3* gene’s coding region [87]. Additionally, Zhao et al. [88] conducted whole-genome resequencing on 206 Hu sheep individuals and identified genes associated with ear size, such as *HOXA1* and *KCNQ2*.

In an effort to safeguard both animals and producers from unintended harm, minimize financial setbacks, and promote animal well-being, initiatives have been undertaken to selectively breed sheep that are naturally hornless. Researchers collected ear tissue samples from 28 ewes of 6 geographically and phenotypically representative Tibetan sheep breeds in Qinghai Province and performed sequencing using the Illumina NovaSeq6000 platform. KEGG pathway analysis revealed that the *RXFP2* gene was significantly enriched in the Relaxin signaling pathway, suggesting its potential association with the hornless phenotype [89]. This also supported the idea that *RXFP2* is a major candidate gene for horn type and size in Chinese native sheep breeds. Sheep horns consist of bone and sheath, and the *BMPR1A* gene is essential for cartilage and osteogenesis differentiation. Therefore, other researchers [90] explored functional loci in the *BMPR1A* gene using whole-genome resequencing and demonstrated that *BMPR1A* was associated with sheep horn type.

### 6.6. Disease Resistance Traits

Brucellosis is a zoonotic disease and a major public health issue [91]. However, the genetic mechanisms underlying brucellosis resistance in sheep remain unclear. Given the urgency of addressing this disease, Li et al. [64] collected 19 brucellosis-resistant sheep and 22 brucellosis-susceptible sheep (BSG) for whole-genome sequencing. They identified nine genes, including *CTNNA3*, *PARD3*, and *PTPRM*, associated with brucellosis susceptibility in sheep. These findings provide valuable molecular markers for breeding brucellosis-resistant sheep.

Additionally, microtia is a congenital malformation of the external ear, with phenotypes ranging from a small auricle to complete absence (anotia). The genetic basis of this condition remains largely unknown, and research on sheep is limited. Understanding the genetic foundation of microtia could aid in the prevention and treatment of this condition. Whole-genome resequencing can help identify high-risk individuals for early intervention and treatment. For example, *HOXA2* has been implicated in developmental deformities by affecting the regulation of the tympanic ring [92]. Furthermore, Mastrangelo et al. [93] used the Illumina OvineSNP50 BeadChip to genotype a total of 40 individuals (20 with microtia and 20 normal individuals) and identified a new candidate gene (*CLRN1*) associated with microtia in sheep. However, the exact function of this gene has not been fully confirmed and requires further detailed investigation. Additionally, researchers genotyped 55 samples from 26 microtia-affected and 29 normal animals and found that the *HMX1* gene had some influence on the development of the external ear in sheep [94].

For most parasitic diseases, they can cause nutritional deficiencies, stunted growth, reduced productivity, and even death in livestock, leading to significant losses [95]. In addition, there are several zoonotic diseases that not only negatively affect animal growth and reproduction but also pose serious risks to human health, threatening public health [96]. The application of whole-genome resequencing technology can reveal the genetic basis of animal diseases, helping scientists understand the biological characteristics, pathogenic mechanisms, and transmission pathways of pathogens. By comparing the genome sequences of different pathogens and hosts, genetic variations associated with diseases can be identified, providing scientific evidence for the development of new diagnostic methods, vaccines, and therapeutic drugs.

### 6.7. Environmental Adaptability Traits

Local breeds often undergo long-term natural selection and adaptation, resulting in unique adaptations to specific environmental conditions [97,98,99]. For instance, in arid regions, local sheep breeds may exhibit higher drought resistance [100,101], allowing them to survive in environments with limited water and feed resources. In mountainous areas, these breeds may possess better cold tolerance and the ability to adapt to high-altitude environments [102,103]. Local sheep breeds can not only contribute significantly to climate mitigation efforts in extreme environments such as arid or high-altitude regions, but these findings also provide valuable genetic resources for future breeding research. They offer new perspectives on how animals adapt to climate change and provide valuable insights for the sustainable development of the sheep industry.

Employed Illumina Ovine SNP50 BeadChip data to study the genetics of KIR sheep from the Pamir Plateau, Qira Black sheep from the Taklamakan Desert, and several introduced breeds: Dorper, Suffolk, and Hu sheep [104]. They identified genes *(ETAA1*, *UBE3D*, *TLE4*, *NXPH1*, *MAT2B*, *PPARGC1A*, *VEGFA*, *TBX15*, and *PLXNA4*) associated with adaptation to plateau, cold, and arid environments. Furthermore, a study [105] utilized medium-density genotyping data from 317 individuals across 15 Welsh sheep breeds, complemented by whole-genome resequencing data from 14 of these breeds. A haplotype-based selection scan of the genotyping data uncovered significant selective sweeps in the *GBA3*, *PPARGC1A*, and *PPP1R16B* genes among highland breeds and in the *RNF24*, *PANK2*, and *MUC15* genes for lowland breeds. This indicates that the observed selective differences in these gene regions may have functional implications, potentially influencing the adaptation of local Welsh sheep to their respective environments. A study on Chinese native sheep also provides valuable insights. Jin et al. [106] analyzed selection traits across 37 Chinese native sheep breeds with diverse climates spanning temperature, altitude, and humidity, discovering genes possibly related to environmental adaptation, including SNP missense mutations in the *TSHR* gene that influence protein structure and stability. Research has shown stronger environmental adaptation to precipitation-related variables than to altitude or temperature-related variables [107]. Subsequently, genes, such as *HIF1AN*, *PDGFA*, *PDGFD*, *ANXA2*, *SOCS2*, *NOXA1*, *WNT7B*, *MMP14*, *GNG2*, *ATF6*, *PGAM2*, *PPP3R1*, *GSTCD*, and *PPARA*, were identified as playing crucial roles in high-altitude adaptation of Tibetan sheep [108].

## 7. Development, Utilization, and Conservation of Genetic Resource Diversity

Currently, 85% of global sheep populations are either endangered or have an unknown status [109]. Local breeds refer to those that have existed in a country or region for a sufficiently long period to genetically adapt to one or more traditional production systems or environments of that country [110]. Biodiversity stems from the continuous selection of unique external phenotypes by humans during successive breeding processes [111]. This process not only shapes biological diversity but also profoundly influences the evolutionary trajectory of species. In agriculture and animal husbandry, humans have gradually formed breeds adapted to different environments and meeting various needs by selecting individuals with specific traits for breeding. Therefore, utilizing whole-genome resequencing to assess the genetic diversity of local livestock breeds is of great significance for their conservation.

Livestock genetic diversity not only influences their production performance but also plays an important role in adapting to environmental changes and disease resistance [112]. Researchers undertook a comprehensive whole-genome resequencing study on 20 Tibetan sheep to investigate the genomic profiles and structural differences between two distinct Tibetan sheep breeds. The study highlighted a set of genes, including *XIRP2*, *ABCB1*, *CA1*, *ASPA*, and *EEF2*, which are believed to influence traits, like body weight, adaptation to the environment, and reproductive efficiency [113]. This research contributes to elucidating the genetic mechanisms behind high-altitude adaptation in Tibetan sheep, offering a wealth of genetic variation data that enhances our knowledge of the genetic traits unique to Tibetan sheep. Similarly, Yang et al. [114] employed whole-genome resequencing to explore the unique germplasm genes, genetic diversity, and genomic structure of inbred, high-yielding Suffolk sheep, uncovering the breed’s distinctive traits and genetic evolution, which lays the groundwork for developing high-yield meat sheep breeds in China. To prevent the increase in inbreeding, the loss of genetic diversity, and the depletion of genetic resources in specific breeds, Gudra et al. [115] performed whole-genome resequencing on 40 local Latvian Blackhead sheep to ensure their conservation. The study found that the contemporary Latvian Blackhead sheep population maintains genetic diversity, providing a basis for future breed management plans. For the critically endangered Italian Quadricorna breed, researchers [116] genotyped 47 individuals and found that the Quadricorna population is similar to breeds originating from the southeastern Mediterranean. Its genetic uniqueness and the inferred systematic geographical reconstruction suggest that the ancient Quadricorna breed exists in the Italian Peninsula.

Livestock and poultry diversity is an important component of biological diversity, and genetic diversity is the core of species diversity. Humanity is a community with a shared future, and the conservation of species’ genetic diversity is crucial for the rational utilization and protection of biological resources, as well as for achieving global sustainable development [117,118]. To improve the theory and methods for the in vivo and ex vivo conservation of sheep germplasm resources, it is essential to establish a dynamic monitoring system for the genetics of germplasm resources and focus on developing an innovative technological system for the conservation and utilization of sheep genetic resources. This involves integrating whole-genome resequencing technology with various preservation techniques, such as somatic cells and stem cells, to establish new, secure methods and technological systems for the conservation of livestock and poultry genetic resources.

## 8. Summary and Outlook

Whole-genome resequencing has facilitated the identification of key genetic variants associated with economically important traits, such as wool quality, lactation performance, reproductive efficiency, disease resistance, and environmental adaptability. The development of high-resolution genetic maps and pan-genome assemblies has further refined our understanding of structural variations and evolutionary mechanisms, supporting more precise genomic selection strategies.

Integrating whole-genome resequencing with transcriptomics, epigenomics, proteomics, and metabolomics is crucial for elucidating the regulatory mechanisms underlying economically important traits in sheep. For example, linking genomic variations to DNA methylation patterns and metabolic pathways can help identify key regulatory networks involved in wool quality, disease resistance, and reproductive efficiency. Additionally, optimizing bioinformatics pipelines for high-throughput variant detection, functional annotation, and interpretation is essential to efficiently handle large-scale sequencing data. Recent advances in CRISPR/Cas9 technology provide a powerful tool for functionally validating candidate genes identified through whole-genome resequencing, enabling precise genetic modifications to enhance economically significant traits. These advances will accelerate the translation of genomes into practical breeding applications, optimize the assessment of genetic diversity and sustainable management of varieties.

## Figures and Tables

**Figure 1 genes-16-00363-f001:**
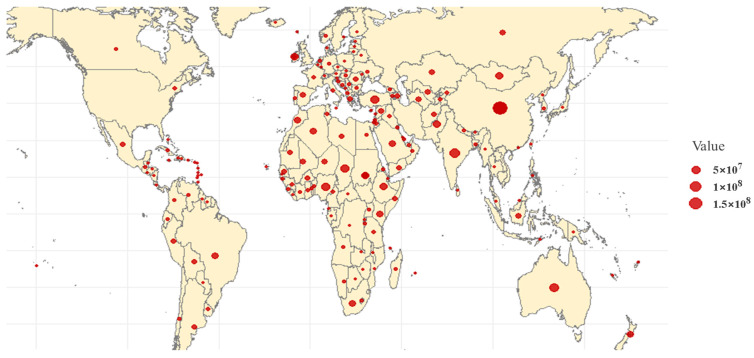
Distribution of global sheep stocks. Note: Figure 1 is based on data from the Food and Agriculture Organization (FAO, http://www.fao.org/faostat/en/#data/QA, accessed on 11 November 2024), illustrating the geographic distribution of sheep populations worldwide.

**Figure 2 genes-16-00363-f002:**
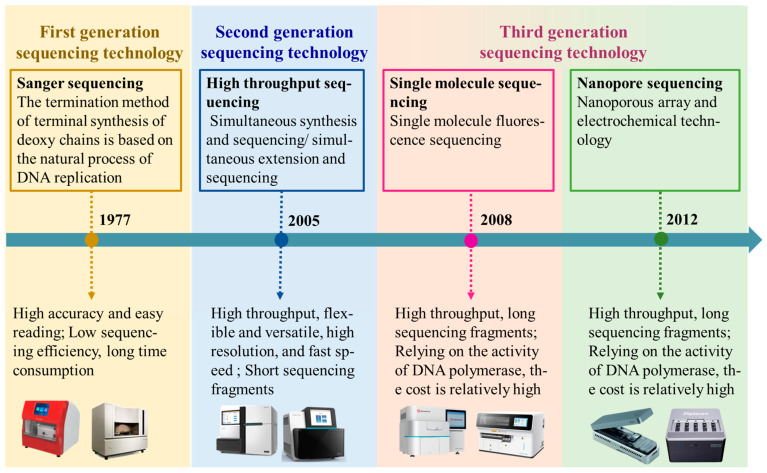
Whole-genome resequencing technology innovation process. Note: The image used in Figure 2 was sourced from the official website of some sequencing companies. The textual content in the figure was conceptualized based on our understanding of sequencing technologies.

**Figure 3 genes-16-00363-f003:**
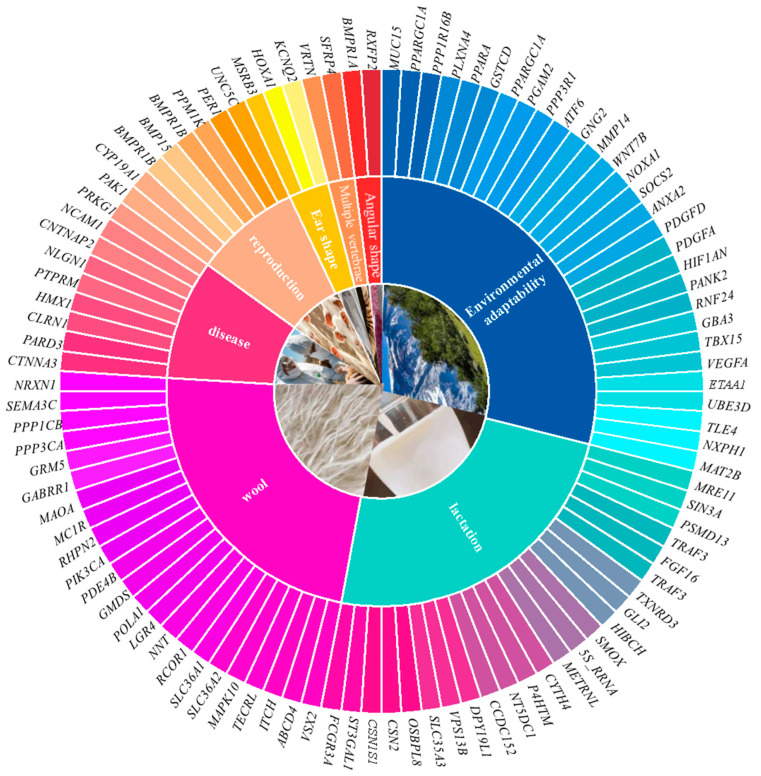
Welcomed presentation of identified genes associated with important traits. Note: Figure 3 is based on the Sheep Trait Association Analysis section of this review, summarizing key genes associated with important traits such as wool quality, milk production, and reproductive performance in sheep.

## Data Availability

No new data were created or analyzed in this study. Data sharing is not applicable to this article.

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
