# Peer review of "Whole-Genome Resequencing in Sheep: Applications in Breeding, Evolution, and Conservation"

_genes, 2025, doi:10.3390/genes16040363_

Round 1

Reviewer 1 Report

Comments and Suggestions for Authors

The authors were focusing on whole genome resequencing publications on sheep.

The development of sheep genetic maps and described in detail, as well as aspects of sequencing technologies. By the way, a question to the authors: On Figure 2, the year 2012 is the closest one to 2025. Were not any mentionable events during that time?

Implications on genome understanding, sheep breeding are fluently described in a followable manner.

Figure 4 is a welcomed presentation of identified genes associated with important traits.

Line 474: 'In summary, a forward-looking analysis is presented in this article'. By definition a review article is a backward investigation of published works. I agree that the MS has a positive tone, it is looking forward, especially at the end of the text, but the MS is a review. It is not forward-looking analysis. Please delete this sentence.

Author Response

Dear Reviewer,

We are grateful for the opportunity to revise our manuscript titled " Genomic Innovations Shaping Sheep Breeding, Evolution, and Conservation " (Manuscript ID: genes-3504392). We appreciate the insightful comments and suggestions provided by the reviewers, which have significantly improved the quality of our work.

In response to the reviewers' feedback, we have made several key revisions to enhance the clarity, specificity, and impact of our manuscript:

Comments 1: The authors were focusing on whole genome resequencing publications on sheep.

Response 1: We appreciate the reviewer's observation. To better reflect the scope of the manuscript, the title has been revised to: “Whole Genome Resequencing in Sheep: Applications in Breeding, Evolution, and Conservation.”

Comments 2: The development of sheep genetic maps and described in detail, as well as aspects of sequencing technologies. By the way, a question to the authors: On Figure 2, the year 2012 is the closest one to 2025. Were not any mentionable events during that time?

Response 2: We thank the reviewer for their comment regarding the timeline in Figure 2. While the figure primarily focuses on the development of sequencing technologies (from first-generation to third-generation sequencing), we acknowledge that the period between 2012 and 2025 may appear sparse in terms of major technological breakthroughs. However, significant progress has been made in the application of whole-genome resequencing (WGR) to sheep genomics during this time, even though no new sequencing technologies have emerged since the introduction of third-generation sequencing (e.g., PacBio and Oxford Nanopore) around 2012.

For example, as mentioned later in the article, the continuous updates to reference genomes and the application of WGR in sheep population genetics and the analysis of economically important traits have been key advancements. These developments have greatly enhanced our understanding of sheep genomics and provided valuable insights for breeding and conservation efforts.

Comments 3: Implications on genome understanding, sheep breeding are fluently described in a followable manner.

Response 3: We thank the reviewer for their positive feedback. To further improve the clarity and flow of the manuscript, we have modified and expanded the relevant descriptions in the text(L67-69, L70-74)

L67-69:These genetic variations form the basis for understanding the genetic architecture of economically important traits in sheep, which directly benefits genomic selection and breeding programs.

L70-74:With the continuous advancement of sequencing technologies and decreasing costs, whole-genome resequencing has become an indispensable tool for not only characterizing genetic diversity and adaptive evolution in sheep populations but also accelerating gnomic selection by enabling the identification of markers linked to eco-nomically important traits.

Comments 4: Figure 4 is a welcomed presentation of identified genes associated with important traits.

Response 4: We appreciate the reviewer's positive comment on Figure 4. To better align with the reviewer's observation, the title of Figure 4 has been revised to: “Welcomed presentation of identified genes associated with important traits.”

Comments 5: Line 474: 'In summary, a forward-looking analysis is presented in this article'. By definition a review article is a backward investigation of published works. I agree that the MS has a positive tone, it is looking forward, especially at the end of the text, but the MS is a review. It is not forward-looking analysis. Please delete this sentence.

Response 5: We thank the reviewer for their constructive feedback. As suggested, we have deleted the sentence: “In summary, a forward-looking analysis is presented in this article.” We agree that the manuscript is primarily a review of existing literature, and we have ensured that the tone remains consistent with this purpose.

Yours sincerely,

Dr. Weidong Deng

Yunnan Provincial Key Laboratory of Animal Nutrition and Feed, Faculty of Animal Science and Technology, Yunnan Agricultural University, Kunming 650201, China

Tel.: +86-871-65220375

Email: dengwd@ynau.edu.cn

Reviewer 2 Report

Comments and Suggestions for Authors

The review focus on an interesting topic, which is the use of whole genome resequencing technologies in sheep genetics, breeding and conservation. The review is comprehensive with the list of 115 relevant references, with most of them (>75 %) published in last 5 years. However, some references were not correctly used, as I listed below.

SImilar study (Woolley et al. doi: 10.1007/s00335-023-10018-z.) was published in 2023 by Woolley et al., but the manuscript is still relevant as it summs up a big portion of later research carried out in this area.

All figures (1-4) need better description to be self-explanatory. The source of data or figure must be stated.

L48-50: Ref [4] (study on Kenya goat breeding goals) doesn’t say much about “molecular biology research and genetic breeding efforts for sheep that have lagged behind those for economically more significant livestock species, such as cattle and pigs“.

L64-65: “This provides a powerful tool for predicting candidate genes for important economic traits in animals and conducting genetic evolution analyses “ [9] – but Ref. [9] deals with whole genome sequencing for molecular genetic diagnosis of rare and unknown human diseases and identification of cancer drivers. However, the review cite a lot of other references, that can support this statement.

L61-95: Authors should differ between whole genome sequencing and whole genome resequencing. With that in mind, authors should edit the whole Chapter 2.

L102-106: [27] refers on the urbanization of Mongolian Plateau, but says nothing about East-West cultural exchanges and dissemination of sheep domestication techniques.

L257-259: Sentence is missing a subject or a reference.

L294-296: Ref. [71] (Dong et al. 2013. Sequencing and automated whole-genome optical mapping of the genome of a domestic goat) says nothing about lambing rate, ovulation rate and their relation to farm production efficiency and economic outcomes.

L307: “we identified 15 candidate genes“    was it your own study?

L448-450: Sentence is missing a subject or a reference.

L485-486: “The lack of a high-quality sheep reference genome is a significant issue“ – such a conclusion is contrary to the Chapter 4. Construction of Sheep Genetic Maps (L143-190).

Author Response

Dear Reviewer,

We are grateful for the opportunity to revise our manuscript titled " Genomic Innovations Shaping Sheep Breeding, Evolution, and Conservation " (Manuscript ID: genes-3504392). We appreciate the insightful comments and suggestions provided by the reviewers, which have significantly improved the quality of our work.

In response to the reviewers' feedback, we have made several key revisions to enhance the clarity, specificity, and impact of our manuscript:

The review focus on an interesting topic, which is the use of whole genome resequencing technologies in sheep genetics, breeding and conservation. The review is comprehensive with the list of 115 relevant references, with most of them (>75 %) published in last 5 years. However, some references were not correctly used, as I listed below.

Comments 1: Similar study (Woolley et al. doi: 10.1007/s00335-023-10018-z.) was published in 2023 by Woolley et al., but the manuscript is still relevant as it summs up a big portion of later research carried out in this area.

Response 1: We thank the reviewer for bringing this study to our attention. The article has now been cited in the text (Lines 96-99): "Over the past two decades, advancements in sequencing technologies have led to the continuous emergence of more detailed genomic maps."* 

This statement is supported by the article "Recent advances in the genomic resources for sheep,"* which states: "The genomic resources for sheep have gradually been improving in quality and resolution over the last twenty years in parallel with advances in sequencing technology. This is particularly evident when describing improvements in the quality and contiguity of the reference genome for sheep."

Comments 2: All figures (1-4) need better description to be self-explanatory. The source of data or figure must be stated.

Response 2: We have added detailed descriptions and sources for all figures:

L41-44

Figure 1. Distribution of global sheep stocks.

Note: Figure 1 is based on data from the Food and Agriculture Organization (FAO, http://www.fao.org/faostat/en/#data/QA), illustrating the geographic distribution of sheep populations worldwide.

L85-87

Figure 2. Whole genome resequencing technology innovation process.

Note: Figure 2 is adapted from references [15] and [16], summarizing the evolution of whole-genome resequencing technologies, including the transition from first-generation Sanger sequencing to third-generation single-molecule real-time sequencing and nanopore sequencing.

L138-141

Figure 3. Distribution and introgression pathways of sheep worldwide [34].

Note: Figure 3 is adapted from reference [34], illustrating the distribution and introgression path-ways of sheep based on the analysis of 154 domesticated breeds and 7 wild relatives of the eight existing sheep species.

L252-255

Figure 4. Main traits of sheep and their corresponding genes.

Note: Figure 4 is based on the Sheep Trait Association Analysis section of this review, summarizing key genes associated with important traits such as wool quality, milk production, and reproductive performance in sheep.

Comments 3: L48-50: Ref [4] (study on Kenya goat breeding goals) doesn’t say much about “molecular biology research and genetic breeding efforts for sheep that have lagged behind those for economically more significant livestock species, such as cattle and pigs“.

Response 3: We have revised the citation and replaced Ref [4] with a more appropriate reference. The sentence now reads (Line 53): 

"The sentence has been cited from the article 'Comprehensive multi-tissue epigenome atlas in sheep: A resource for complex traits, domestication, and breeding,' which mentions: 'However, the currently available annotations of regulatory elements in the sheep genome are limited to only a few tissues, and the comprehensive annotation of functional elements in the sheep genome is lagging behind that of other animal species, such as pigs, cattle, chickens, and model organisms. This limited annotation hinders our understanding of the molecular mechanisms underlying complex agronomic traits in sheep.'"

Comments 4: L64-65: “This provides a powerful tool for predicting candidate genes for important economic traits in animals and conducting genetic evolution analyses “ [9] – but Ref. [9] deals with whole genome sequencing for molecular genetic diagnosis of rare and unknown human diseases and identification of cancer drivers. However, the review cite a lot of other references, that can support this statement.

Response 4: We have replaced Ref [9] with a more relevant citation. The revised sentence now reads (Line 68):

The sentence has been cited from the article "Whole-genome resequencing reveals loci under selection during chicken domestication, " which mentions: "Whole-genome resequencing of wild and domestic chickens has enabled the identification of genetic variations that are associated with domestication traits, providing insights into the molecular basis of these economically important characteristics. "

Comments 5: L61-95: Authors should differ between whole genome sequencing and whole genome resequencing. With that in mind, authors should edit the whole Chapter 2.

Response 5: We have added a clear distinction between whole-genome sequencing (WGS) and whole-genome resequencing (WGR) in Lines 91-93:

"Unlike whole-genome sequencing, which involves sequencing a genome for the first time to create a reference sequence, whole-genome resequencing focuses on sequencing multiple individuals within a species and aligning their sequences to an existing reference genome."

This explanation is supported by citations from the articles "Coming of age: ten years of next-generation sequencing technologies" and "Whole Genome Resequencing Helps Study Important Traits in Chickens."

Comments 6: L102-106: [27] refers on the urbanization of Mongolian Plateau, but says nothing about East-West cultural exchanges and dissemination of sheep domestication techniques.

Response 6: We have replaced Ref [27] with a more appropriate citation. The revised sentence  now reads(Line 109):

The sentence has been cited from the article "Mitogenomic Meta-Analysis Identifies Two Phases of Migration in the History of Eastern Eurasian Sheep, " which mentions: "The Mongolian Plateau region was a secondary zone/center of sheep dispersal as a “transportation hub” in eastern Eurasia: Sheep from the Middle Eastern domestication center were inferred to have migrated through Caucasus and Central Asia and arrived in North and Southwest China."

Comments 7: L257-259: Sentence is missing a subject or a reference.

Response 7: We have revised the sentence and added the subject (Lines 273-276):

"Cao et al. [63] analyzed the population structure, genetic diversity, and selection traits of Romanov sheep using whole-genome sequencing data from 114 individuals, including 17 Romanov sheep and 10 other European breeds, and identified MC1R as a candidate gene associated with the breed's unique gray wool color."

Comments 8: L294-296: Ref. [71] (Dong et al. 2013. Sequencing and automated whole-genome optical mapping of the genome of a domestic goat) says nothing about lambing rate, ovulation rate and their relation to farm production efficiency and economic outcomes.

Response 8: We have replaced Ref [71] with a more relevant citation. The revised sentence now reads (Line 312):

The sentence has been cited from the article "Impact of production strategies and animal performance on economic values of dairy sheep traits, " which mentions: "Results of the sensitivity analysis are presented in detail for the four economically most important traits: 150 days milk yield, conception rate of ewes, litter size, and ewe productive lifetime."

Comments 9: L307: “we identified 15 candidate genes“    was it your own study?

Response9: L323: We have clarified this by changing "we" to "they" (Line 323):

"They identified 15 candidate genes."

Comments 10: L448-450: Sentence is missing a subject or a reference.

Response 10: We have revised the sentence and added the subject (Lines 448-450):

"Similarly, Yang et al. [111] employed whole-genome resequencing to explore the unique germplasm genes, genetic diversity, and genomic structure of inbred, high-yielding Suffolk sheep, uncovering the breed's distinctive traits and genetic evolution, which lays the groundwork for developing high-yield meat sheep breeds in China."

Comments 11: L485-486: “The lack of a high-quality sheep reference genome is a significant issue“ – such a conclusion is contrary to the Chapter 4. Construction of Sheep Genetic Maps (L143-190).

Response 11: We agree with the reviewer and have deleted the contentious passage, as it was inconsistent with the content of Chapter 4. The removed text included:

"The lack of a high-quality sheep reference genome is a significant issue. A high-quality reference genome is crucial for accurately identifying and aligning the genes of sheep, and it is more representative and scientifically valid. To address this problem, the latest sequencing technologies and bioinformatics methods can be employed to construct a high-quality sheep reference genome, followed by genome sequencing of different sheep breeds to enhance the representativeness and diversity of the reference genome. Future applications of third-generation sequencing and pangenome analysis will further refine our understanding of genetic diversity in sheep. Additionally, integrating multi-omics approaches, including epigenomics and transcriptomics, may provide deeper insights into adaptive traits and selective breeding strategies."

Yours sincerely,

Dr. Weidong Deng

Yunnan Provincial Key Laboratory of Animal Nutrition and Feed, Faculty of Animal Science and Technology, Yunnan Agricultural University, Kunming 650201, China

Tel.: +86-871-65220375

Email: dengwd@ynau.edu.cn
